# Determining respiratory rate from photoplethysmogram and electrocardiogram signals using respiratory quality indices and neural networks

**Stephanie Baker** [1]*, **Wei Xiang** [2], **Ian Atkinson** [3]

**1** College of Science and Engineering, James Cook University, Cairns, Queensland, Australia, **2** School of Engineering and Mathematical Sciences, La Trobe University, Melbourne, Victoria, Australia, **3** eResearch Centre, James Cook University, Townsville, Queensland, Australia

* stephanie.baker@jcu.edu.au

**Data Availability Statement:** The data used for this paper is from the Medical Information Mart for Intensive Care (MIMIC) database. In particular, the data is from the MIMIC waveform database, which is an open access database accessible at https://

## Abstract

Continuous and non-invasive respiratory rate (RR) monitoring would significantly improve patient outcomes. Currently, RR is under-recorded in clinical environments and is often measured by manually counting breaths. In this work, we investigate the use of respiratory signal quality quantification and several neural network (NN) structures for improved RR estimation. We extract respiratory modulation signals from the electrocardiogram (ECG) and photoplethysmogram (PPG) signals, and calculate a possible RR from each extracted signal. We develop a straightforward and efficient respiratory quality index (RQI) scheme that determines the quality of each moonddulation-extracted respiration signal. We then develop NNs for the estimation of RR, using estimated RRs and their corresponding quality index as input features. We determine that calculating RQIs for modulation-extracted RRs decreased the mean absolute error (MAE) of our NNs by up to 38.17%. When trained and tested using 60-sec waveform segments, the proposed scheme achieved an MAE of 0.638 breaths per minute. Based on these results, our scheme could be readily implemented into non-invasive wearable devices for continuous RR measurement in many healthcare applications.

## Introduction

Respiratory rate (RR) is a fundamental physiological parameter, and abnormality in this vital sign is one of the earliest indicators of critical illness. One recent study found that elevated respiratory rate was a key predictor of clinical deterioration within 48 hours of discharge from the emergency department [1]. Another classical study determined that the occurrence of at least one RR $\geq$ 27 breaths per minute (BrPM) in a 72 hour period was a strong predictor of cardiac arrest [2]. Elevated RR has also been linked to increased mortality [3], while relative changes in RR have been shown to indicate patient stability [4]. In children, elevated RR is a primary indicator of pneumonia, an infection that is the most common cause of death in

doi.org/10.13026/c2607m The code for this work
is available at: https://github.com/stephb23/
RespiratoryRate.

**Funding:** This work was supported by the
Australian Government Research Training Program
Scholarship.

**Competing interests:** The authors have declared
that no competing interests exist.

children aged 0-5 [5, 6]. Clearly, abnormalities or variations in the RR are key indicators of
clinical deterioration.

Despite the clinical significance of RR, several studies have noted that it is historically less
recorded than other vital signs [1, 7–9]. This has somewhat improved with the introduction
of the Modified Early Warning Score [7], which incorporates measurement of RR. However,
one study observed that nurses still don't measure RR in 50% of cases [9]. Time constraints
and the lack of equipment for measuring RR were both cited as reasons for not monitoring
this parameter.

This lack of recording can be partially attributed to the fact that there is a lack of tools avail-
able for automatically measuring RR. Currently, most methods for automatic RR measure-
ment rely on oronasal systems incorporating sensors including capnography, temperature,
and moisture sensors [5]. However, these have not been widely adopted, with issues related to
cost, wearability, and accuracy identified for existing automated devices [5].

Manual measurement remains the accepted method for determining RR. To obtain RR, it is
recommended that healthcare staff count the number of breaths a patient takes over a one-
minute period [6]. However, several studies have found that both doctors and nurses estimate
respiratory rate over shorter time periods, or without counting the breath at all [10, 11]. Accu-
racy of manual RR calculations can be affected by patient awareness [9], as well as time con-
straints, interruptions from patients and other staff, and patient agitation [5, 11].

In addition to the complications associated with obtaining an accurate manual RR mea-
surement, there is also a significant time cost. One study found that as much as 7.2% of nurses'
time was spent performing patient assessment, including measurement of RR [12]. There are
approximately 3 million registered nurses in America, earning an average of $75,510 USD per
annum each as of May 2018 [13]. Thus, the total financial cost incurred by time nurses spend
on patient assessment exceeds 16 billion USD per year.

Given the major limitations in measuring RR, it is clear that a reliable method of automatic
and continuous monitoring of this vital sign in a non-invasive manner would significantly
improve patient outcomes in hospitals. Additionally, given the usefulness of RR as an early
indicator of critical illness, continuous at-home measurement of RR could be lifesaving for at-
risk patients living alone.

Several recent studies have investigated the use of photoplethysmogram (PPG) and electro-
cardiogram (ECG) signals to derive RR in a wearable and non-invasive manner [14–19]. Res-
piration modulates the ECG and PPG signals in three main ways—baseline wander (BW)
modulation, amplitude modulation (AM) and respiratory sinus arrhythmia (RSA) modula-
tion, more commonly known as frequency modulation (FM). These modulations are caused
by movement associated with breathing, and various responses to the change in intrathoracic
pressure during respiration [20].

In order to accurately estimate RR, several recent studies have developed respiratory quality
indices (RQIs) to determine which of the extracted modulations are of the highest quality [17,
18, 21]. This in turn allows for identification of which modulation-extracted RRs are realistic,
thus allowing for more accurate estimation of actual RR.

Interestingly, there are very few studies that have attempted to estimate RR from PPG and
ECG using machine learning (ML). The best performing ML-enabled technique was presented
in [17], where a mean absolute error (MAE) of 0.71 BrPM was achieved using linear regres-
sion. While these are reasonably good results, we will demonstrate that they can be improved
upon by instead using neural networks (NNs) in combination with our own novel RQI
scheme.

In this work, we develop an RQI scheme for assessing the quality of modulation-extraction
respiration signals. The proposed scheme uses statistics regarding the signal variation to assign

'good' or 'bad' ratings to RRs calculated from modulation-extracted signals. We train and test multiple neural networks, comparing the performance in two scenarios: one where only RR features are used as features, and the other where both RR and corresponding RQIs are used.

The remainder of this paper is structured as follows. Section II describes the methodology utilized for obtaining signal quality and an overall RR estimation using various NN structures. Section III presents results and discussion before Section IV concludes the paper and provides recommendations for future work. Acronyms and abbreviations used throughout this work are defined in Table 1 below for the convenience of the reader.

## Methodology

### Obtaining data

Data for this work was obtained from the open-source Medical Information Mart for Intensive Care (MIMIC-III) database [22], which features an extremely large number of records from patients admitted to intensive care units (ICUs) between 2001-2012. Data used to conduct this research was first accessed in 2019. To train the neural networks, ECG and PPG signals were needed to derive RR from the BW, AM, and FM modulations. Additionally, a reference "true" RR signal was needed to provide the neural networks with an expected output RR. As such, the PhysioBank ATM tool [23] was used to obtain a list of all records containing ECG, PPG, and respiratory waveforms from the MIMIC-III database. Then, a Python script was developed to download all relevant records as MATLAB-compatible files, utilizing several functions from

**Table 1. Acronyms and abbreviations.**

| Abbreviation | Definition |
| --- | --- |
| AM | Amplitude modulation |
| BiLSTM | Bidirectional long short-term memory |
| BrPM | Breaths per minute |
| BrTBr | Breath-to-breath |
| BTB | Beat-to-beat |
| BW | Baseline wander |
| DCV | Differential coefficient of variation |
| ECG | Electrocardiogram |
| FM | Frequency modulation |
| HR | Heart rate |
| ICU | Intensive care unit |
| LOA | Limit of agreement |
| LSTM | Long short-term memory |
| MAE | Mean absolute error |
| MD | Mean difference |
| MIMIC | Medical Information Mart for Intensive Care |
| ML | Machine learning |
| NN | Neural network |
| PCC | Pearson's correlation coefficient |
| PPG | Photoplethysmograph |
| RMSE | Root mean square error |
| RQI | Respiratory quality index |
| RR | Respiratory rate |
| RSA | Respiratory sinus arrhythmia |
| SQI | Signal quality index |

the Waveform Database Toolbox [24]. After running this script, a total of 8,781 records were obtained. No exclusions were made based on patient demographics, diagnoses or treatments received, as we aimed to develop an all-inclusive scheme that could measure respiratory rate irrespective of whether respiration was being affected by health conditions or respiratory support treatments. Patient demographics are also not attached to many of the waveform records used, however an overview of patient demographics across the entire MIMIC-III database is presented in the original paper describing the database [22].

## Preprocessing data

The primary preprocessing performed was the denoising of ECG and PPG signals. Many of the ECG and PPG signals were affected by baseline wander that could be attributed both to respiration and other movement. Thus, baseline wander was removed from each signal using a low-pass Chebyshev filter and stored for later use.

After removing the low-frequency BW components from the signals, it was observed that many ECG signals still appeared noisy. To denoise the ECG signals, a seventh-order Savitsky-Golay filter was utilized. This filter type was chosen due as they are well-known to preserve small details of a waveform, such as the Q- and S-waves found in ECG signals.

After signals were denoised, all records including ECG, PPG and respiratory signals were split into segments. In this work, we trialled three different segment lengths to determine the most suitable length for accurate RR prediction. The segments chosen were 20, 30, and 60 seconds. These segment lengths are commonly used in the literature, allowing for fair comparison. They also each enable very frequent RR estimation, while also providing a wide enough window to accurately calculate even very low RRs. At this point, any segment with a missing signal or flat-lining signal was discarded.

For each record segment, the R-waves (or peaks) of the ECG signals were found, as well as the peaks of the PPG and reference RR signals. Additionally, the beat-to-beat intervals were calculated for PPG and ECG signals, and the breath-to-breath (BrTBr) interval was calculated for RR signals. Heart rate (HR) was then calculated from both the PPG and ECG signals, before RR was calculated from the reference respiration signal. This extracted information was then used by our purpose-built signal quality index (SQI) as described in the next section, to determine the overall quality of the segment and thus the segment's suitability for training and testing the neural networks.

Furthermore, the RR of each segment was calculated by finding the average period between peaks of the respiration signal. This period represents one full breath, and thus the RR was calculated using the following formula:

$$RR_{\text{true}} = \frac{60}{mean(BrTBr_1, BrTBr_2, \ldots, BrTBr_n)} \tag{1}$$

where 'BrTBr' represents a breath-to-breath interval measured in seconds, 'n' is the number of BrTBr intervals within the respiratory signal segment, and the 'RR$_{\text{true}}$' is taken as the "true RR" for that segment.

## Signal quality assessment

Signal quality assessment is vital to ensure that neural networks are learning from realistic data. One significant work [25] found that simple conditional statements can be used to effectively assess the quality of PPG, ECG, and blood pressure signals. In these works, various sanity checks were performed to determine the quality of a signal, such as ensuring that heart rate

(HR) and beat-to-beat (BTB) intervals were within reasonable ranges. Reasonable range for RR were determined based on clinical medicine resources

In this work, PPG and ECG signals are considered with respect to calculating RR, and as such the quality of the respiration signal is also vital. As such, this work develops an SQI tool based on conditional statements relevant to the problem in order to successfully classify a record containing PPG, ECG and respiration signals as either "good" or "bad" based on a series of conditional statements. This is described by the following algorithm:

**Algorithm 1 Signal quality index algorithm**

```
Input: hr_ppg, hr_ecg, ppg_peak_ratio, ecg_peak_ratio, ppg_btb_ratio,
ecg_btb_ratio, true_rr, true_rr_peak_ratio, true_rr_brtbr_ratio
Output: signal_quality
1: if [(abs(hr_ppg—hr_ecg) < 10) & (hr_ppg > 40) & (hr_ppg < 180) &
(ppg_peak_ratio < 1.5) &
  (ecg_peak_ratio < 1.5) & (ptp_btb_ratio < 1.5) &
  (ecg_btb_ratio < 1.5) & (true_rr > 8) &
  (true_rr < 35) & (true_rr_peak_ratio < 1.5)
  & (true_rr_brtbr_ratio < 1.5)] then
2:     signal_quality = 1
3: else
4:     signal_quality = 0
5: else if
```

In this algorithm, *hr_ppg* and *hr_ecg* are the HR values calculated from the PPG and ECG signals, respectively. They are compared to each other to verify that they were acceptably similar, then *hr_ppg* was checked to ensure that HR was within the physiologically probable range of 40-180 bpm [26]. Meanwhile, *ppg_peak_ratio, ecg_peak_ratio* and *true_rr_peak_ratio* represent the ratio of the maximum to minimum peak heights for the PPG, ECG and reference RR signals respectively, and *ppg_btb_ratio, ecg_btb_ratio* and *true_rr_brtbr_ratio* represent the ratio of maximum to minimum PPG signal BTB intervals, ECG signal BTB intervals and reference RR signal BrTBr intervals respectively. It was checked that each of these ratios was <1.5 to ensure that there was acceptable consistency within each individual signal, as consistency is a strong indicator of signal quality. Lastly, *true_rr* represents the RR extracted from the reference signal using Eq 1, and it was checked that this fell within the conservative range of 8-35, substantially broader than the 15-30 BrPM defined as normal RR [27]. Records that met all criteria were assigned a *signal_quality* of 1, meaning "good", while failure to meet any criteria resulted in a *signal_quality* of 0, or "bad".

After testing all segments with the SQI tool, there were 19,084 "good" 20-second segments, 7,301 "good" 30-second segments, and 1,300 "good" 60-second segments for use in training and testing the model. The next stage was to extract features from each of these signals for use in training the neural networks. This was a multi-step process, which begins with the extraction of respiration-induced modulations from the ECG and PPG signal as discussed in the following subsection.

## Extracting respiratory signals from ECG and PPG

Respiration can modulate the ECG and PPG signals in three key ways—baseline wander (BW) modulation, amplitude modulation (AM) and frequency modulation (FM) caused by respiratory sinus arrhythmia. These modulations are shown in comparison to signals unaffected by respiration (without modulation) in Fig 1. As previously discussed, one or more respiratory modulations may be absent from the PPG and ECG signals of some patients. As such, endeavouring to extract all three key modulations from both the ECG and PPG signal will greatly enhance a neural network's ability to estimate true RR.

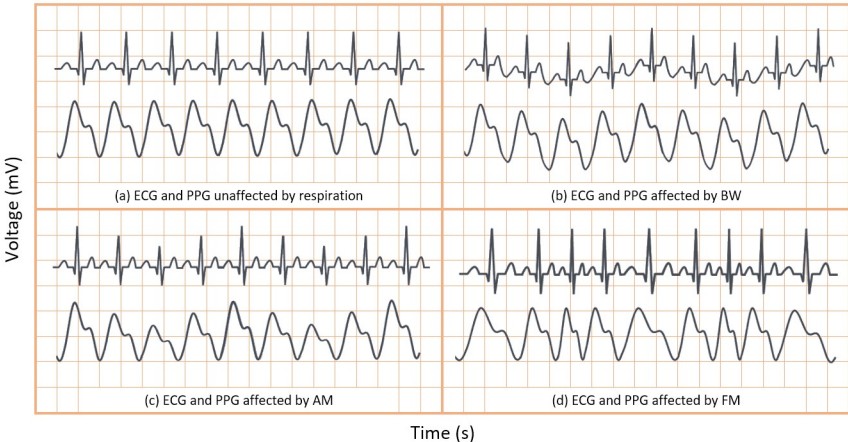

**Fig 1. Sample ECG and PPG signals with and without effects of respiratory modulation.** (A) unaffected by respiratory modulation. (B) affected by baseline wander. (C) affected by amplitude modulation. (D) affected by frequency modulation.

**Extracting respiratory signals.** In the context of respiration, BW is the overall shift in the baseline of an ECG or PPG signal due to respiration, as is shown in Fig 1B. BW was obtained by low-pass filtering the ECG and PPG signals. Hereafter the BW signals extracted from the PPG and ECG signals are denoted as PPG-BW and ECG-BW, respectively. Meanwhile, AM presents as the variation in peak heights in the ECG and PPG signals, after BW has been removed, as shown in Fig 1C. Finally, FM presents in ECG or PPG signals as varying beat duration, as shown in Fig 1D. Thus, AM and FM respiration signals are easily derived from the peak heights and BTB intervals of the waveforms, respectively. The AM and FM signals extracted from PPG and ECG are henceforth denoted as PPG-AM, PPG-FM, ECG-AM, and ECG-FM.

After BW, AM and FM signals were extracted from the PPG and ECG signal, peaks and troughs of each signal were calculated and stored in six separate vectors. Breath-to-breath intervals, as well as the intervals between trough locations, were also calculated and stored in six additional vectors. These parameters were then used by the developed respiratory quality index (RQI) tool described in the following subsection.

Finally, a possible respiratory rate was derived from each signal by finding the average period between peaks (the breath-to-breath interval), and thus determining the number of breaths per minute. This process is mathematically defined as:

$$RR_{\text{signal}} = \frac{60}{mean(BrTBr_1, BrTBr_2, \ldots BrTBr_n)} \tag{2}$$

where 'BrTBr' is a breath-to-breath interval, 'n' is the number of BrTBr intervals within the extracted signal, and the 'signal' of $RR_{\text{signal}}$ is the PPG-BW, PPG-AM, PPG-FM, ECG-BW, ECG-AM or ECG-FM.

Overviews of the distribution of modulation-derived respiratory rates, along with the distribution of true respiratory rates, are presented for the 20-second segment dataset in S1 Table, the 30-second segment dataset in S2 Table, and the 60-second segment dataset in S3 Table.

## Respiratory quality assessment

The development of an RQI scheme that assigns each modulation-extracted respiratory signal a quality rating on some scale could improve RR estimation algorithms, as knowledge about the quality of each estimated RR can enhance the networks ability to determine true RR based.

In this work, we propose a efficient and effective RQI scheme that considers the variance in peak heights (ph), trough depths (td), and the distances between peak pairs (p-p) and trough pairs (t-t) for any given extracted RR signal.

Consistency is a key indicator of respiratory signal quality, and as such we propose the differential coefficient of variation (DCV) metric, a variation on the the coefficient of variation, to quantify how much variation is in the signal. We calculate the DCV for peak heights, trough depths, peak-to-peak distances and trough-to-trough distances as follows:

$$DCV = 1 - \frac{\sigma}{\mu} \tag{3}$$

where $\sigma$ represents the standard deviation (SD) and $\mu$ represents the mean of the vector of data. Then, we calculate the DCV for the four properties of interest—peak height, trough depths, distance between peak pairs, and distance between trough pairs. These are denoted as $DCV_{ph}$, $DCV_{td}$, $DCV_{p-p}$ and $DCV_{t-t}$ in Eq (4), respectively.

As is shown in Eq (4), we then find the average of the four DCVs. In Eq (3), most values will fall between 0-1, but there is a possibility of negative values where there is no consistency. The **max** calculation in the following equation is used to ensure that the resulting RQI-C value falls between 0 and 1, even in the highly unlikely case where there is no consistency in any of the DCVs.

$$RQI = \max(\sum \frac{DCV_{ph} + DCV_{td} + DCV_{p-p} + DCV_{t-t}}{4}, 0) \tag{4}$$

The calculated RQI will be 0 in the case where there is no consistency, and 1 in the case where there is perfect consistency. As consistency is the best indicator of signal quality, higher RQI values indicate better quality signals.

This scheme was used to calculate an RQI for each of the six modulation-extracted respiratory signals in every record; PPG-BW, ECG-BW, PPG-AM, ECG-AM, PPG-FM, and ECG-FM.

## Feature selection

We developed two separate feature vectors to analyse the performance of neural networks with and without the RQI features as inputs. For the first test, we selected solely the modulation-extracted RRs, resulting in a six-feature input vector as follows:

$$[RR_{ECG-BW}, RR_{PPG-BW}, RR_{ECG-AM},$$
$$RR_{PPG-AM}, RR_{ECG-FM}, RR_{PPG-FM}]$$

For the second test, we created a feature vector that included RQIs calculated using our proposed scheme, along with the modulation-extracted RRs. The resultant twelve-feature vector is as follows:

$$[RQI_{ECG-BW}, RR_{ECG-BW}, RQI_{PPG-BW}, RR_{PPG-BW},$$
$$RQI_{ECG-AM}, RR_{ECG-AM}, RQI_{PPG-AM}, RR_{PPG-AM},$$
$$RQI_{ECG-FM}, RR_{ECG-FM}, RQI_{PPG-FM}, RR_{PPG-FM}, ]$$

These two feature vectors were constructed for every record that was classified as 'good' by the SQI tool.

## Neural network structure

In this work, we use a bidirectional long short-term memory (BiLSTM) network structure to predict respiratory rate from the input features. BiLSTM cells are updated using the same mathematical structure as unidirectional long short-term memory cells, but the data is passed through the network both as-is (forwards) and in reversed order (backwards). The results of these operations is then concatenated before passing to the next layer. The mathematical structure of a single forward or backwards pass is described by the following equations, with interested readers referred to the original paper that introduced LSTM for further details regarding mathematical theory [28].

$$\tilde{c}_t = \tanh(w_c[a_{(t-1)}, x_t] + b_c) \tag{5}$$

$$f_t = \sigma(w_f[a_{(t-1)}, x_t] + b_f) \tag{6}$$

$$u_t = \sigma(w_u[a_{(t-1)}, x_t] + b_u) \tag{7}$$

$$o_t = \sigma(w_o[a_{(t-1)}, x_t] + b_o) \tag{8}$$

$$c_t = u_t \bullet \tilde{c}_t + f_t \bullet c_{(t-1)} \tag{9}$$

$$a_t = o_t \bullet \tanh(c_t) \tag{10}$$

where $w_c$, $w_f$, $w_u$ and $w_o$ refer to the learned weights for their respective operations, while $b_c$, $b_f$, $b_u$ and $b_o$ are the learned biases. These are learnt during training using the Adam optimization algorithm [29], a common optimization algorithm that uses adaptive learning rates and momentum to converge quickly and efficiently on the true optimal solution. Additionally, the parameter $a_{(t-1)}$ refers to the output of the previous layer, while $x_t$ is the input for timestep $t$. Eq (9) utilizes the results of Eqs (5)–(7) as well as the cell state of the previous time step, $c_{(t-1)}$ to update the cell state, and Eq (10) uses the resultant $c_c$ as well as the output gate results. The '•' symbol in Eqs (9) and (10) represents element-wise matrix multiplication, and the function $\sigma()$ in Eqs (6)–(8) is the sigmoid activation function, which is defined as $\sigma(z) = \frac{1}{1+e^{-z}}$.

The neural network structure utilised in this work includes three hidden BiLSTM layers each comprised of the forward and backwards passes followed by the concatenation operation. The first two hidden layers return a sequence of all hidden cell states, hence the high number of concatenation operations. The third hidden layer outputs only the final state of each cell from both the forward and backwards pass, and these are then concatenated. The network structure is illustrated in Fig 2 below.

The NN structure included 128 hidden units per hidden layer and a batch size of 1024 to enable good generalization without overfitting. The aforementioned Adam optimization [29] function is used to update weights and biases during training, while the mean absolute error (MAE) is used as the loss function.

## Training and testing

In this work, the NN structure was trained and tested six times to compare the performance of the network using the six different feature vectors, as follows:

- All 12 features, as calculated from 20-second segments

- The 6 RR features only, as calculated from 20-second segments

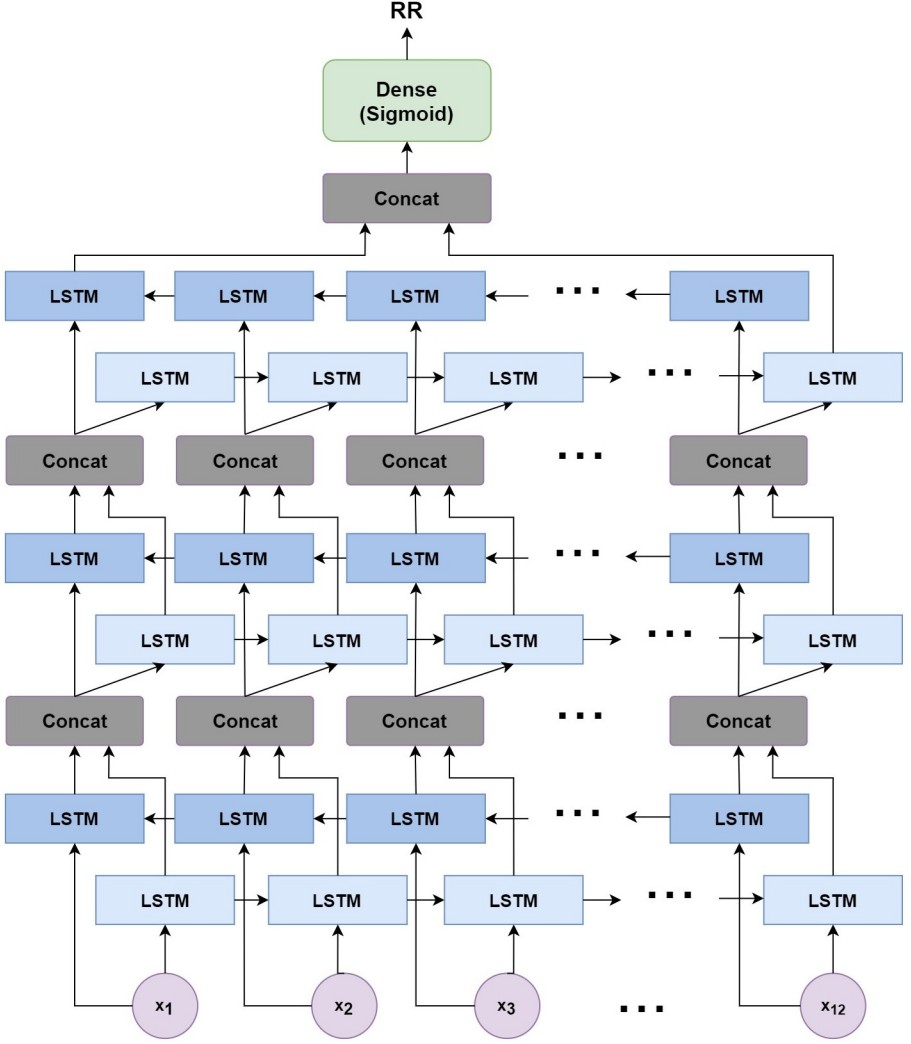

**Fig 2. Structure of the BiLSTM model.**

- All 12 features, as calculated from 30-second segments

- The 6 RR features only, as calculated from 30-second segments

- All 12 features, as calculated from 60-second segments

- The 6 RR features only, as calculated from 60-second segments

The data was pseudorandomly shuffled before being split into subsets for training, validating, and testing. 80% of the data was used for training the NNs, 10% was used for fine-tuning hyperparameters through the validation process, and the remaining 10% of unseen data was utilized to fairly test the models.

## Results and discussion

After training and testing all of the NN configurations, statistical and graphical analysis was conducted to assess the performance of each network. In terms of statistical analysis, several informative metrics were considered: mean absolute error (MAE), root mean square error

(RMSE), and Pearson's correlation coefficient (PCC). Furthermore, Bland Altman analysis was conducted by calculating the bias or mean difference (MD), the difference or width between the limits of agreement (LOAs), and the percentage of results (of mean vs. difference between true and predicted RRs) that fall between said LOAs.

MAE gives key insight into how skilled the network is at producing a reasonable prediction for RR. RMSE is indicative of how many high-range errors there are, and thus provides information about whether the network has fit appropriately to the data. PCC indicates the level of linear correlation, and will give a result between 0 and ±1, representing no correlation and total positive/negative correlation respectively.

In terms of the Bland Altman analysis metrics, a low MD along with narrow LOAs is a good indicator of strong agreement between the two methods of measurement. In Bland Altman analysis, each data point is the result of comparing the mean of the two measurement methods with the difference between their predictions. If the majority of these results fall within the LOAs, then this further indicates a strong level of agreement between the two measurements. As such, a high-performing network would have low MD, low LOA width, and a high percentage of results within the LOAs.

The results of calculating these metrics for the BiLSTM NNs trained using each feature vector are shown in Table 2. These results clearly indicate that the inclusion of RQIs calculated using our proposed scheme greatly improves the success of machine learning in estimating true RR. Table 2 shows that the inclusion of RQI features reduced the MAE by up to 36.89% when compared to the equivalent networks that were trained using solely the modulation-extracted RRs. Significant improvements RMSE and PCC are all also visible across all NN structures considered. In all cases, including RQI features increased the level of agreement between true and predicted RR measurements, narrowing the LOA width. MDs were extremely small across all networks.

Table 2 also shows that the BiLSTM network model performs strongly regardless of the segment length used to derive the RR and RQIs, however MAE is shown to decrease as segment length is increased. The overall lowest MAE was 0.638, achieved by the network trained on RRs & RQIs extracted from 60 second segments. As the inclusion of RQI features is shown to reduce MAE, the remainder of our analysis will focus on the networks trained with both RR & RQI features.

To further analyse the predictive performance of the BiLSTM network, the error histograms in Fig 3 were created to graphically investigate the spread of errors in RR predictions. To create these figures, all errors were rounded to the nearest 0.25 to allow for better visualisation. These figures reiterate the high accuracy of the systems trained using both RR and RQI features.

We further analyse the performance of the BiLSTM network when trained on records with different segment lengths via the Bland Altman plots in Fig 4. Bland Altman plots are used to assess the level of agreement between two measurement methods—in this case, we compare the predictions made by our proposed BiLSTM model against the reference RR measurement

**Table 2. Performance of BiLSTM NN using various feature vectors for estimating respiratory rate.**

| Segment Length | Features | MAE (BrPM) | RMSE (BrPM) | PCC | MD | LOA Width | % in LOAs |
|---|---|---|---|---|---|---|---|
| 20 seconds | RR & RQIs | 0.821 | 2.236 | 0.891 | -0.08 | 8.76 | 95.44% |
| | RR Only | 1.301 | 2.776 | 0.829 | -0.16 | 10.87 | 92.77% |
| 30 seconds | RR & RQIs | 0.747 | 1.926 | 0.901 | 0.14 | 7.54 | 95.21% |
| | RR Only | 1.116 | 2.430 | 0.839 | -0.04 | 9.53 | 93.43% |
| 60 seconds | RR & RQIs | 0.638 | 1.575 | 0.932 | -0.15 | 6.17 | 95.38% |
| | RR Only | 0.711 | 1.731 | 0.919 | -0.14 | 6.79 | 96.15% |

(a)

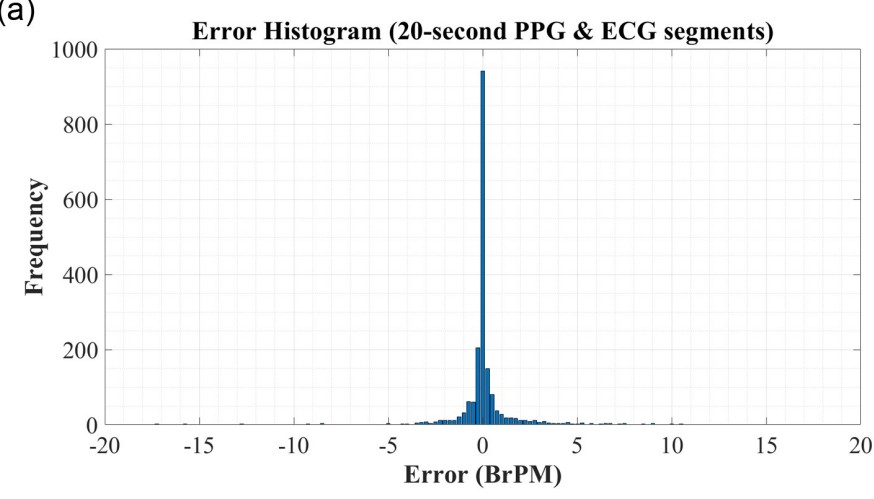

(b)

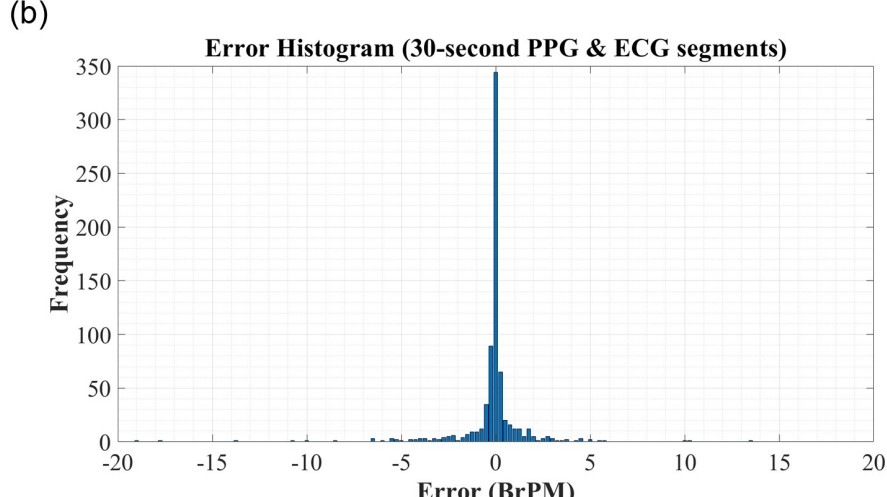

(c)

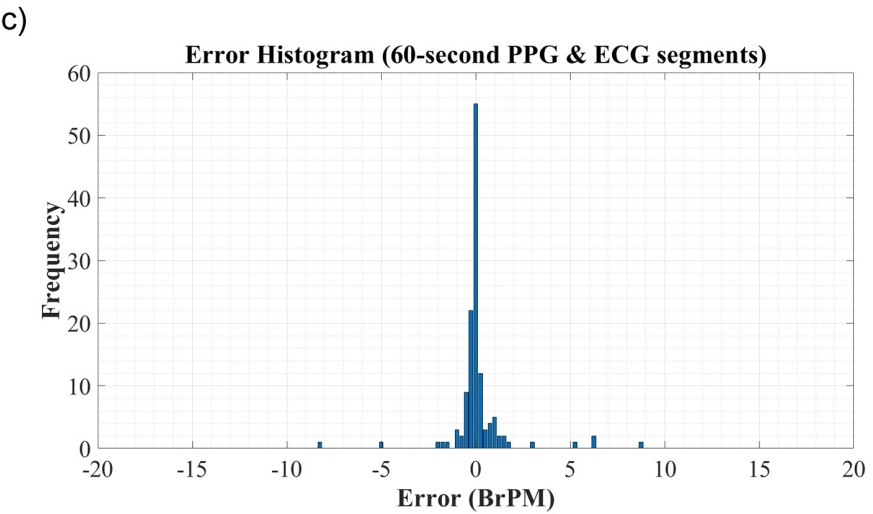

**Fig 3.** Error Histograms for RR Estimation using RR & RQI features derived from: (A) 20-second PPG & ECG segments. (B) 30-second PPG & ECG segments. (C) 60-second PPG & ECG segments.

from the MIMIC-III database. The difference between the two measurements is plotted against the mean of the two measurements, and as such a high density around the central 'mean difference' line within the 'limits of agreement' indicates strong agreement between two schemes. In each plot, the difference vs. mean results were often extremely close together and appeared to overlap. As such, density color scales are included in Fig 4 to better illustrate the concentration of points. As can be seen from this plot, there is a high density of points along the mean difference line, with 95.44%, 95.21%, and 95.38% of results falling within the limits of agreement for the models utilising features extracted from 20-second, 30-second, and 60-second ECG and PPG segments, respectively. This indicates a strong correlation between the true RRs and those predicted by our proposed network, regardless of the segment length used for feature extraction.

To further assess the correlation between the true and predicted values for RR, the regression plots in Fig 5 were constructed. In each figure, the thick black line represents what 'perfect' correlation would look like, while the dashed black line is the actual correlation achieved by the network. From this regression plot, it is clear that there is a strong correlation between the predictions made by the BiLSTM model and the reference RRs obtained from the MIMIC-III database, regardless of the segment length used to derive the features. In each plot, the actual correlation line falls very close to the ideal correlation line, and very few data points are outliers in the trend.

Overall, the proposed BiLSTM model shows low error and a high level of agreement with gold-standard measurement, regardless of which segment length is used for feature extraction. Performance increased as segment length increased, but even shorter segments showed strong results. In all cases, the inclusion of features calculated based on our proposed RQI scheme greatly improves the performance of the BiLSTM neural network. Therefore, it is clear that a BiLSTM model utilising extracted RRs and our proposed RQIs would significantly improve RR calculation in clinical and at-home environments, with longer ECG and PPG segments for feature extraction leading to the most accurate predictions.

## Comparison to previous works

The results obtained by our BiLSTM models compare well to previous works when the feature vectors with both modulation-extracted RRs and corresponding RQIs were used, regardless of the segment length that these features were extracted from. This is shown in Table 3. It is clear that the proposed model outperforms the previous state-of-the-art schemes for RR estimation from ECG and PPG signals, achieving significantly better MAE and comparable RMSE. Unfortunately, PCC was not provided by previous works in Table 3 so could not be considered when making comparisons to the literature.

Compared to the works presented in Table 3, our BiLSTM models with RR and RQI features perform extremely strongly regardless of segment length used to extract these parameters. The RMSEs of all models were lower than the previous works in the literature. In terms of MAE, the model trained using 60s segments outperformed all previous works. One work [17] reported a lower MAE of 0.71 BrPM on the Capnobase database than was achieved by our models based on 20s and 30s signal segments, however the MAE of [17] rose to 3.12 BrPM when the scheme was applied to the larger and more comprehensive MIMIC database. As our work is based on MIMIC data, the latter result is more comparable. Overall, our LSTM models both outperform the literature in terms of MAE.

Interestingly, our enhanced results were achieved even where the short window length of 20 seconds was used. Accuracy increased with time, however the risk of artefacts impacting

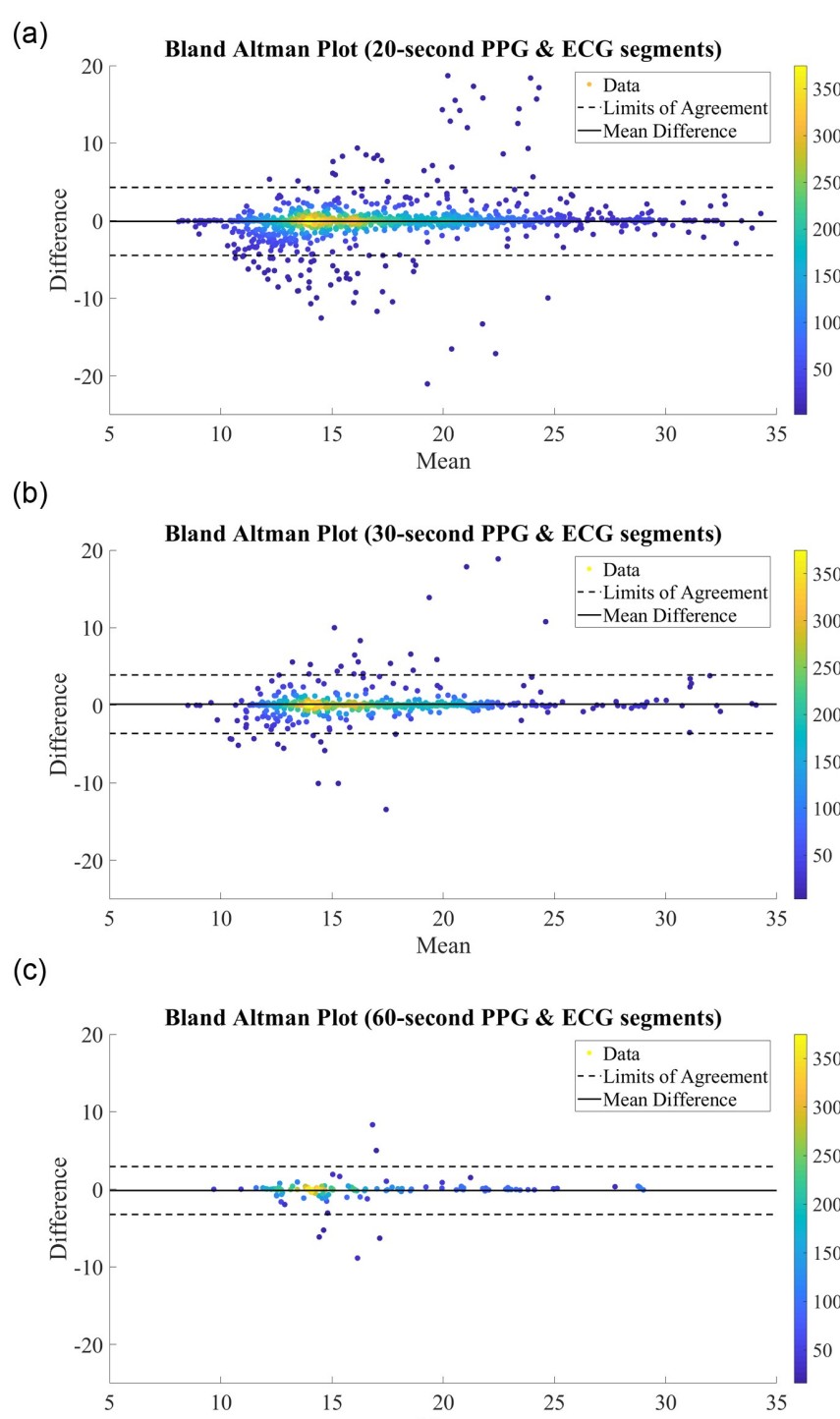

**Fig 4.** Bland Altman Plots for RR Estimation using RR & RQI features derived from (A) 20-second PPG & ECG segments. (B) 30-second PPG & ECG segments. (C) 60-second PPG & ECG segments.

(a)

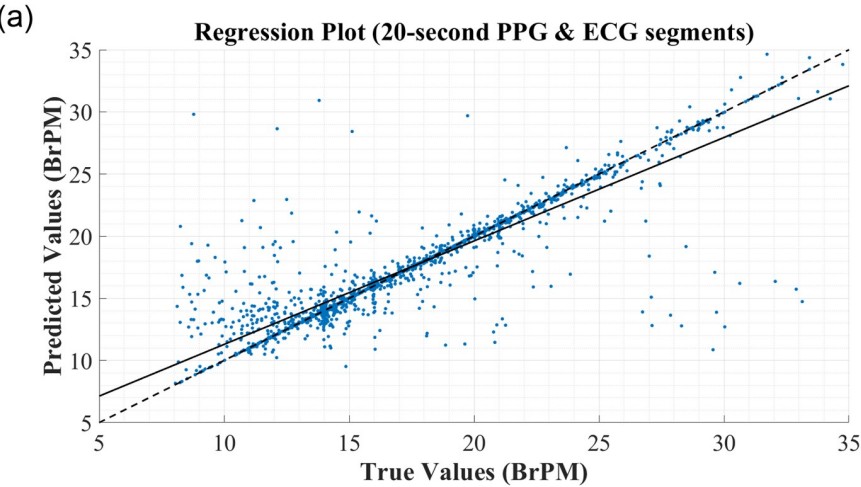

(b)

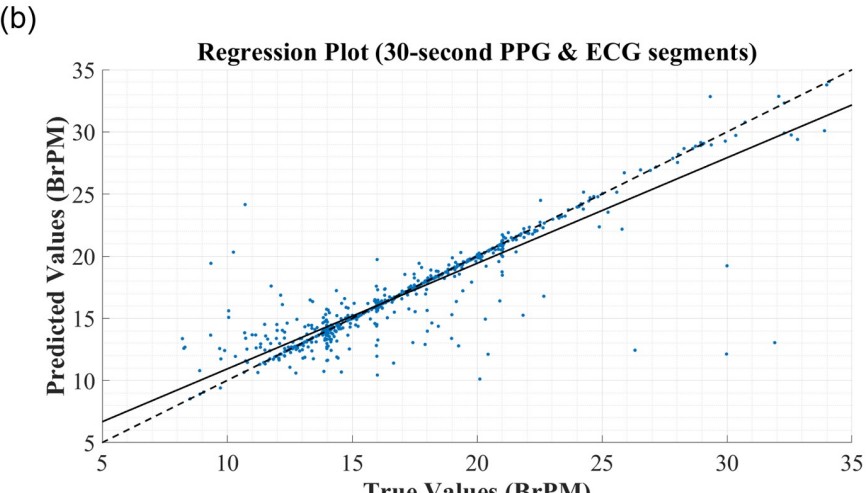

(c)

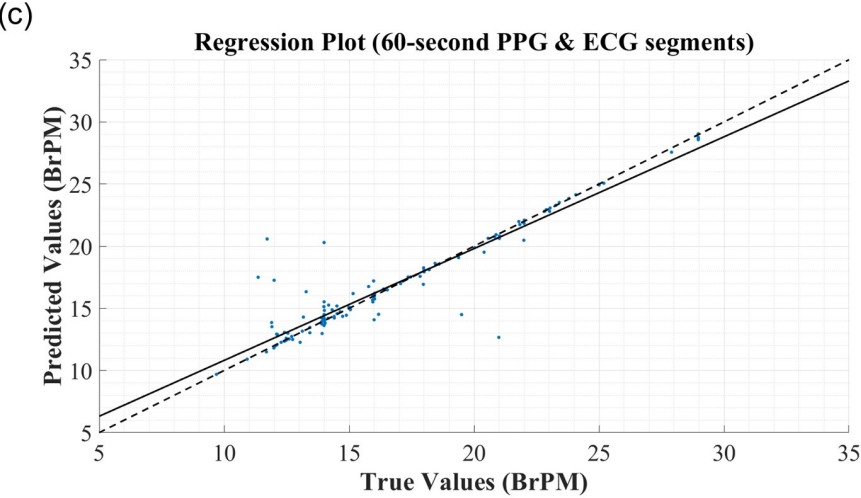

**Fig 5.** Regression Plots for RR Estimation using RR & RQI features derived from: (A) 20-second PPG & ECG segments. (B) 30-second PPG & ECG segments. (C) 60-second PPG & ECG segments.

**Table 3. Comparison to previous works.**

| | | Error Metrics (BrPM) | |
|---|---|---|---|
| | Segment Length (s) | MAE (BrPM) | RMSE (BrPM) |
| Orphanidou [14] | 60 | 1.80 | N/A |
| Karlen [15] | 60 | N/A | 2.3 |
| Birrenkott [17] | 32 | 0.71[1], 3.12[2] | N/A |
| Pirhonen [19] | N/A | 1.764 | 3.996 |
| BiLSTM + RQI | 20 | 0.821 | 2.236 |
| BiLSTM + RQI | 30 | 0.747 | 1.926 |
| BiLSTM + RQI | 60 | 0.638 | 1.575 |

[1] Based on testing against 42 Capnobase [30] records

[2] Based on testing against 53 records MIMIC-II [31] records

[3] Based on testing against 42 Capnobase [30] records, results varied based on window length selected and on signal used (PPG or ECG)

the signal quality also increases with the length of the segment. This suggests that our scheme could predict RR faster, while also achieving a lower error.

It is also worth noting that the previous works largely relied on very small datasets. Through using a large database for this work, it has been possible to thoroughly validate the performance of the network across a large and diverse set of patients. Our results were obtained through testing our scheme on 1,909 segments, compared to other recent works such as [17, 19] where 95 and 29 records were used to obtain the results in Table 3, respectively. This ultimately means that our network is more likely to translate to real-world application with success, while many of the previous works would need to be validated on larger databases.

## Conclusion

In this work, an RQI scheme was developed to enhance the performance of neural networks utilizing the respiratory modulations of ECG and PPG signals to estimate true RR. The proposed RQI scheme was implemented and tested to evaluate improvements in the performance of NNs in predicting RR from modulation-extracted RR estimates, with exceptional results.

When RQIs were used alongside modulation-extracted RRs as input features, a bidirectional LSTM model was able to achieve the low MAE of 0.821 BrPM. This is a significant improvement when compared to other works in the literature, and proves that RQIs can greatly enhance the performance of neural networks.

With further validation on non-ICU data, this scheme would likely be suitable for at-home healthcare monitoring due to the wearable nature of PPG and ECG sensors. In our future work, we will investigate this application.

The results of this paper show that a device implementing our proposed RQI scheme with a BiLSTM NN would be suitable for continuous and non-invasive monitoring of respiratory rate, using hardware that is already in place in many healthcare environments. We suggest that this algorithm would be suitable for clinical use. With further validation on persons outside of ICU, it would also be suitable for at-home health monitoring. This scheme could greatly improve early prediction of potentially fatal conditions, enhance remote healthcare, and ultimately improve patient outcomes.

## Supporting information

**S1 Table. Respiratory rate distribution in 20-second segment dataset.**
(PDF)

**S2 Table. Respiratory rate distribution in 30-second segment dataset.**
(PDF)

**S3 Table. Respiratory rate distribution in 60-second segment dataset.**
(PDF)

## Author Contributions

**Conceptualization:** Stephanie Baker.

**Investigation:** Stephanie Baker.

**Methodology:** Stephanie Baker.

**Software:** Stephanie Baker.

**Supervision:** Wei Xiang, Ian Atkinson.

**Writing – original draft:** Stephanie Baker.

**Writing – review & editing:** Stephanie Baker, Wei Xiang, Ian Atkinson.

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
