## [Decision Letter · Decision Letter 0]

22 Feb 2021

PONE-D-21-02517

Determining respiratory rate from photoplethysmogram and electrocardiogram signals using respiratory quality indices and neural networks

PLOS ONE

Dear Dr. Baker,

Thank you for submitting your manuscript to PLOS ONE. After careful consideration, we feel that it has merit but does not fully meet PLOS ONE’s publication criteria as it currently stands. Therefore, we invite you to submit a revised version of the manuscript that addresses the points raised during the review process.

We look forward to receiving your revised manuscript.

Kind regards,

Chi-Hua Chen, Ph.D.

Academic Editor

PLOS ONE

Journal Requirements:

2. Thank you for providing the date(s) when patient medical information was initially recorded. Please also include the date(s) on which your research team accessed the databases/records to obtain the retrospective data used in your study.

3. To meet our data availability requrements, please provide the data used for this study. This may be a supplementary table that includes basic demographic information and measurements of the six modulation-extracted respiratory signals.

4.Please provide a summary table of patient demographics.

5.Thank you for stating the following in the Acknowledgments Section of your manuscript:

"This work was supported by the Australian Government Research Training Program

Scholarship."

"The authors received no specific funding for this work."

Reviewers' comments:

Reviewer's Responses to Questions

**Comments to the Author**

1. Is the manuscript technically sound, and do the data support the conclusions?

Reviewer #1: Partly

2. Has the statistical analysis been performed appropriately and rigorously? 

Reviewer #1: I Don't Know

3. Have the authors made all data underlying the findings in their manuscript fully available?

Reviewer #1: No

4. Is the manuscript presented in an intelligible fashion and written in standard English?

Reviewer #1: Yes

5. Review Comments to the Author

Reviewer #1: In “Determining respiratory rate from photoplethysmogram and electrocardiogram signals using respiratory quality indices and neural networks” by Baker, Xiang, and Atkinson, the authors propose a method to approximate Respiratory Rate (RR) from other common signals (ECG, PPG) and based on idealized neural network models fit (using ‘Adam Optimization’).

A key result appears to be in Table 1 where including RQI lowers errors and increases correlation. I appreciate Table 2, comparison to previous works. This paper overall appears sound. My biggest concerns with lack of clarity, basic definitions of functions (see below), and a seeming lack of care or experience in thinking through figures are disappointing. But hopefully these issues can be fixed.

1) Consistent with PLoS policy, the authors should make all of the code for their models freely available at a public repository.

They should also have the scripts to generate the exact figures and not just the minimal working model.

This to me is the most crucial part of having this paper accepted because aspects of the models that are not easily understood with the current descriptions (see below).

2) All of the acronyms are quite hard to follow. I strongly urge the authors to add a table upfront to define the many acronyms/abbreviations.

3) Eq (5)—(10): what is the function \\sigma()? Or is this \\sigma just the std. dev from eq (3)? I understand tanh() is commonly used to represent a sigmoidal nonlinear saturation, a commonly used transfer function in neural networks. Also if the variable c_t are matrices, what are the state variables? Also I’m not quite familiar with Adam Optimization… the extent of my experience in deep neural nets is that the large # of params begs to use

standard gradient decent. I and other readers would greatly appreciate it if the authors please explain these details and say a few words about the

optimization algorithm.

4) Figures 1–4 need axes title, labels, units, numbers. I’ve never seen anything so bare unless it was part of a schematic that is part of a larger figure.

For readability, the authors should consider making these all into 1 figure with labels. In its current form, it is really hard to see anything they are trying to communicate.

5) Same comments for Figs 6–8, 9–11, 12–14: please combine the figures; this would make it easier for readers to digest your results.

6) In Table 2: presumably the other methods [14—19] did not have PCC? If so, that would important to include; if not, please state this.

Minor: MAE is not defined in the abstract but all other acronyms are. Mean absolute error defined on line 57. Please define what it is or don’t use the abbreviation.

6. PLOS authors have the option to publish the peer review history of their article (what does this mean?). If published, this will include your full peer review and any attached files.

Reviewer #1: No

---

## [Author Response · Author response to Decision Letter 0]

23 Mar 2021

Reviewer 1: Thank you for your detailed and insightful review of our manuscript. Your feedback was very helpful. We have incorporated all of your suggestions into our revision.

---

## [Decision Letter · Decision Letter 1]

26 Mar 2021

Determining respiratory rate from photoplethysmogram and electrocardiogram signals using respiratory quality indices and neural networks

PONE-D-21-02517R1

Dear Dr. Baker,

We’re pleased to inform you that your manuscript has been judged scientifically suitable for publication and will be formally accepted for publication once it meets all outstanding technical requirements.

Kind regards,

Chi-Hua Chen, Ph.D.

Academic Editor

PLOS ONE

Additional Editor Comments (optional):

Reviewers' comments:

Reviewer's Responses to Questions

**Comments to the Author**

1. If the authors have adequately addressed your comments raised in a previous round of review and you feel that this manuscript is now acceptable for publication, you may indicate that here to bypass the “Comments to the Author” section, enter your conflict of interest statement in the “Confidential to Editor” section, and submit your "Accept" recommendation.

Reviewer #1: All comments have been addressed

2. Is the manuscript technically sound, and do the data support the conclusions?

Reviewer #1: Yes

3. Has the statistical analysis been performed appropriately and rigorously? 

Reviewer #1: Yes

4. Have the authors made all data underlying the findings in their manuscript fully available?

Reviewer #1: Yes

5. Is the manuscript presented in an intelligible fashion and written in standard English?

Reviewer #1: Yes

6. Review Comments to the Author

Reviewer #1: Thanks for making the request edits. From what I can tell, the research looks technically sound now.

7. PLOS authors have the option to publish the peer review history of their article (what does this mean?). If published, this will include your full peer review and any attached files.

Reviewer #1: No

---

## [Editor Report · Acceptance letter]

30 Mar 2021

PONE-D-21-02517R1 

Determining respiratory rate from photoplethysmogram and electrocardiogram signals using respiratory quality indices and neural networks 

Dear Dr. Baker:

I'm pleased to inform you that your manuscript has been deemed suitable for publication in PLOS ONE. Congratulations! Your manuscript is now with our production department. 

Kind regards, 

on behalf of

Professor Chi-Hua Chen 

Academic Editor

PLOS ONE